# Unsupervised Attention-guided Image-to-Image Translation

**Youssef A. Mejjati**
University of Bath
yam28@bath.ac.uk

**Christian Richardt**
University of Bath
christian@richardt.name

**James Tompkin**
Brown University
james_tompkin@brown.edu

**Darren Cosker**
University of Bath
D.P.Cosker@bath.ac.uk

**Kwang In Kim**
University of Bath
k.kim@bath.ac.uk

## Abstract

Current unsupervised image-to-image translation techniques struggle to focus their attention on individual objects without altering the background or the way multiple objects interact within a scene. Motivated by the important role of attention in human perception, we tackle this limitation by introducing unsupervised attention mechanisms that are jointly adversarially trained with the generators and discriminators. We demonstrate qualitatively and quantitatively that our approach attends to relevant regions in the image without requiring supervision, which creates more realistic mappings when compared to those of recent approaches.

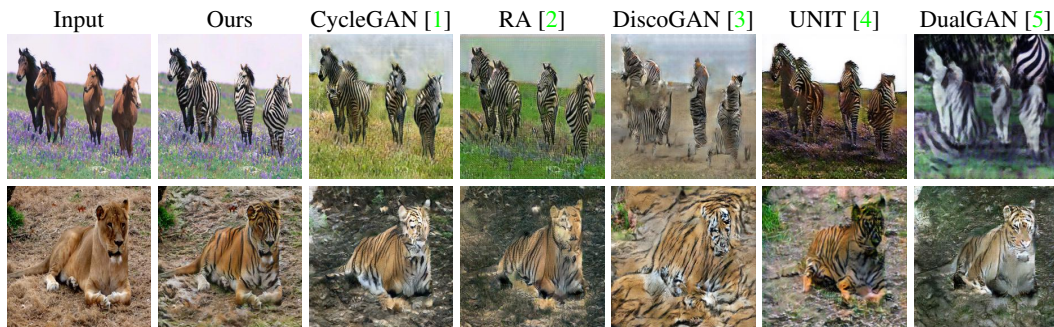

Figure 1: By explicitly modeling attention, our algorithm is able to better alter the object of interest in unsupervised image-to-image translation tasks, without changing the background at the same time.

## 1   Introduction

Image-to-image translation is the task of mapping an image from a source domain to a target domain. Applications include image colorization [6], image super-resolution [7, 8], style transfer [9], domain adaptation [10] and data augmentation [11]. Many approaches require data from each domain to be paired or under alignment, e.g., when translating satellite images to topographic maps, which restricts applications and may not even be possible for some domains. Unsupervised approaches, such as DiscoGAN [3] and CycleGAN [1], overcome this problem with cyclic losses which encourage the translated domain to be faithfully reconstructed when mapped back to the original domain.

Existing algorithms feed an input image to an encoder–decoder-like neural network architecture called the *generator*, which tries to translate the image. Then, this output is fed to a *discriminator*

which attempts to classify if the output image has indeed been translated. In these generative adversarial networks (GANs), the quality of the generated images improves as the generator and discriminator compete to reach the Nash equilibrium expressed by the minimax loss of the training procedure [12].

However, these approaches are limited by the system's inability to attend only to specific scene objects. In the unsupervised case, where images are not paired or aligned, the network must additionally learn which parts of the scene are intended to be translated. For instance, in Figure 1, a convincing translation between the horse and zebra domains requires the network to attend to each animal and change only those parts of the image. This is challenging for existing approaches, even if they use a localized loss like PatchGAN [13], as the network itself has no explicit attention mechanism. Instead, they typically aim to minimize the divergence between the underlying data-generating distribution for the entire image in the source and target domains. To overcome this limitation, we propose to minimize the divergence between only the relevant parts of the data-generating distributions for the source and target domains. For this, we find inspiration from attentional mechanisms in human perception [14], and their successful application in machine learning [2, 15]. We add an attention network to each generator in the CycleGAN setup. These are jointly trained to produce attention maps for regions that the discriminator 'considers' are the most discriminative between the source and target domains. Then, these maps are applied to the input of the generator to constrain it to relevant image regions. The whole network is trained end-to-end with no additional supervision. We qualitatively and quantitatively show that explicitly incorporating attention into image translation networks significantly improves the quality of translated images (see Figure 1).

## 2 Related work

**Image-to-image translation.** Contemporary image-to-image translation approaches leverage the powerful ability of deep neural networks to build meaningful representations. Specifically, GANs have proven to be the gold standard in achieving appealing image-to-image translation results. For instance, Isola et al.'s pix2pix algorithm [9] uses a GAN conditioned on the source image and imposes an $L_1$ loss between the generated image and its ground-truth map. This requires the existence of ground-truth paired images from each of the source and target domains. Zhu et al.'s unpaired image-to-image translation network [1] builds upon pix2pix and removes the paired input data burden by imposing that each image should be reconstructed correctly when translated twice, i.e., when mapped from source to target to source. These maps must conserve the overall structure and content of the image. DiscoGAN [3] and DualGAN [5] use the same principle, but with different losses, making them more or less robust to changes in shape.

Some unsupervised translation approaches assume the existence of a shared latent space between source and target domains. Liu and Tuzel's Coupled GAN (CoGAN) [16] learns an estimate of the joint data-generating distribution using samples from the marginals, by enforcing source and target discriminators and generators to share parameters in low-level layers. Liu et al.'s unsupervised image-to-image translation networks (UNIT) [4] build upon Coupled GAN by assuming the existence of a shared low-dimensional latent space between the source and target domains. Once the image is mapped to its latent representation, then a generator decodes it into its target domain version. Huang et al.'s multi-modal UNIT (MUNIT) [17] framework extends this idea to multi-modal image-to-image translation by assuming two latent representations: one for 'style' and one for 'content'. Then, the cross-domain image translation is performed by combining different content and style representations.

Given input images depicting objects at multiple scales, the aforementioned approaches are sometimes able to translate the foreground. However, they generally also affect the background in unwanted ways, leading to unrealistic translations. We demonstrate that our algorithm is able to overcome this limitation by incorporating attention into the image translation framework.

Attending to specific regions within image translation has recently been explored by Ma et al. [18], who attempt to decouple local textures from holistic shapes by attending to local objects of interest (e.g., eyes, nose, and mouth in a face); this is manifested through attention maps as individual square image regions. This limits the approach, as (1) it assumes that all objects are the same size, corresponding to the sizes of the square attention maps, and (2) it involves tuning hyper-parameters for the number and size of the square regions. As a consequence, this approach cannot straightforwardly deal with image translation without altering the background.

**Attention learning.** Attention learning has benefited from advances in deep learning. Contemporary approaches use convolution-deconvolution networks trained on ground-truth masks [19], and combine these architectures with recurrent attention models. Specifically, Kuen et al.'s saliency

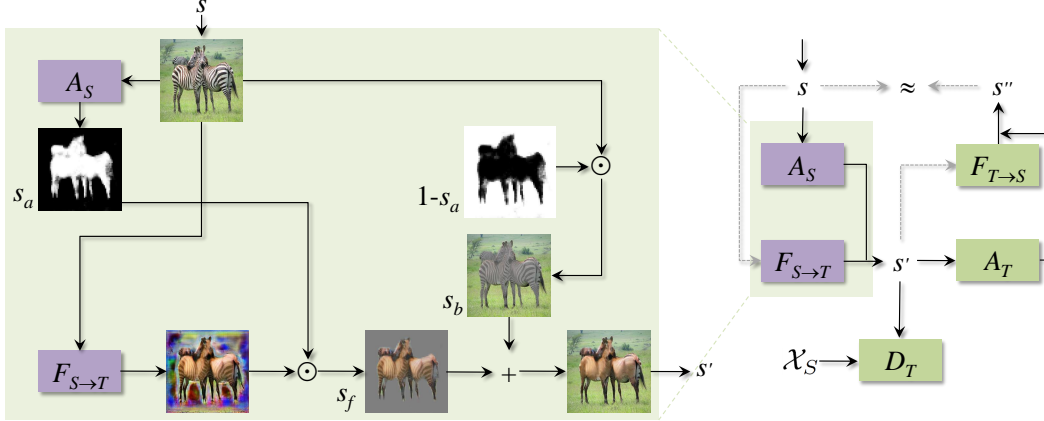

Figure 2: Data-flow diagram from the source domain $S$ to the target domain $T$ during training. The roles of $S$ and $T$ are symmetric in our network, so that data also flows in the opposite direction $T \rightarrow S$.

detection [20] uses Recurrent Neural Networks (RNN) to adaptively select a sequence of local regions in the input image for saliency estimation. Then, these local estimates are combined into a global estimate. Such approaches cannot be applied in our setting, since they require supervision.

Unsupervised attention learning includes Mnih et al.'s recurrent model of visual attention [15], which uses only a few learned square regions of the image trained from classification labels. This approach is not differentiable and requires training with reinforcement learning, which is not straightforward to apply in our problem. More recently, attention has been enforced on activation functions to select only task-relevant features [2, 21]. However, we show in experiments that our approach of enforcing attention on the input image provides better results for image-to-image translation.

Learning attention also encourages the generation of more realistic images compared to classic vanilla GANs. For example, Zhang et al.'s self-attention GANs [22] constrain the generator to gradually consider non-local relationships in the feature space by using unsupervised attention, which produces globally realistic images. Yang et al.'s recursive approach [23] generates images by decoupling the generation of the foreground and background in a sequential manner; however, its extension to image-to-image translation is not straightforward as in that case we only care about modifying the foreground. Attention has also been used for video generation [24], where a binary mask is learned to distinguish between dynamic and static regions in each frame of a generated video. The generated masks are trained to detect unrealistic motions and patterns in the generated frames, whereas our attention network is trained to find the most discriminative regions which characterize a given image domain. Finally, Chen et al.'s contemporaneous work shares our goal of learning an attention map for image translation [25]; we will discuss the differences between our methods after explaining our approach (see Section 4).

## 3  Our approach

The goal of image translation is to estimate a map $F_{S \rightarrow T}$ from a source image domain $S$ to a target image domain $T$ based on independently sampled data instances $\mathcal{X}_S$ and $\mathcal{X}_T$, such that the distribution of the mapped instances $F_{S \rightarrow T}(\mathcal{X}_S)$ matches the probability distribution $\mathrm{P}_T$ of the target. Our starting point is Zhu et al.'s CycleGAN approach [1], which also learns a domain inverse $F_{T \rightarrow S}$ to enforce *cycle consistency*: $F_{T \rightarrow S}(F_{S \rightarrow T}(\mathcal{X}_S)) \approx \mathcal{X}_S$. The training of the transfer network $F_{S \rightarrow T}$ requires a discriminator $D_T$ to try to detect the translated outputs from the observed instances $\mathcal{X}_T$. For cycle consistency, the inverse map $F_{T \rightarrow S}$ and the corresponding discriminator $D_S$ are simultaneously trained.

Solving this problem requires solving two equally important tasks: (1) locating the areas to translate in each image, and (2) applying the right translation to the located areas. We achieve this by adding two attention networks $A_S$ and $A_T$, which select areas to translate by maximizing the probability that the discriminator makes a mistake. We denote $A_S : S \rightarrow S_a$ and $A_T : T \rightarrow T_a$, where $S_a$ and $T_a$ are the attention maps induced from $S$ and $T$, respectively. Each attention map contains per-pixel $[0,1]$ estimates. After feeding the input image to the generator, we apply the learned mask to the generated image using an element-wise product '$\odot$', and then add the background using the

inverse of the mask applied to the input image. As such, $A_S$ and $A_T$ are trained in tandem with the generators; Figure 2 visualizes this process.

Henceforth, we will describe only the map $F_{S \to T}$; the inverse map $F_{T \to S}$ is defined similarly.

## 3.1 Attention-guided generator

First, we feed the input image $s \in S$ into the generator $F_{S \to T}$, which maps $s$ to the target domain $T$. Then, the same input is fed to the attention network $A_S$, resulting in the attention map $s_a = A_S(s)$. To create the 'foreground' object $s_f \in T$, we apply $s_a$ to $F_{S \to T}(s)$ via an element-wise product on each RGB channel: $s_f = s_a \odot F_{S \to T}(s)$ (Figure 2 shows an example). Finally, we create the 'background' image $s_b = (1 - s_a) \odot s$, and add it to the masked output of the generator $F_{S \to T}$. Thus, the mapped image $s'$ is obtained by:

$$s' = \underbrace{s_a \odot F_{S \to T}(s)}_{\text{Foreground}} + \underbrace{(1 - s_a) \odot s}_{\text{Background}}. \tag{1}$$

**Attention map intuition.** The attention network $A_S$ plays a key role in Equation 1. If the attention map $s_a$ was replaced by all ones, to mark the entire image as relevant, then we obtain CycleGAN as a special case of our approach. If $s_a$ was all zeros, then the generated image would be identical to the input image due to the background term in Equation 1, and the discriminator would never be fooled by the generator. If $s_a$ attends to an image region without a relevant foreground instance to translate, then the result $s'$ will preserve its source domain class (i.e. a horse will remain a horse).

In other words, the image parts which most describe the domain will remain unchanged, which makes it straightforward for the discriminator $D_T$ to detect the image as a fake. Therefore, the only way to find an *equilibrium* between generator $F_{S \to T}$, attention map $A_S$, and discriminator $D_T$ is for $A_S$ to focus on the objects or areas that the corresponding discriminator *thinks* are the most descriptive within its domain (i.e., the horses). The discriminator mechanism which makes GAN generators produce realistic images *also* makes our attention networks find the domain-descriptive objects in the images.

The attention map is continuous between $[0,1]$, i.e., it is a matte rather than a segmentation mask. This is valuable for three reasons: (1) it makes estimating the attention maps differentiable, and so able to train at all, (2) it allows the network to be uncertain about attention during the training process, which allows convergence, and (3) it allows the network to learn how to compose edges, which otherwise might make the foreground object look 'stuck on' or produce fringing artifacts.

**Loss function.** This process is governed by the adversarial energy:

$$\mathcal{L}_{adv}^s(F_{S \to T}, A_S, D_T) = \mathbb{E}_{t \sim P_T(t)}\big[\log(D_T(t))\big] + \mathbb{E}_{s \sim P_S(s)}\big[\log(1 - D_T(s'))\big]. \tag{2}$$

In addition, and similarly to CycleGAN, we add a cycle-consistency loss to the overall framework by enforcing a one-to-one mapping between $s$ and the output of its inverse mapping $s''$:

$$\mathcal{L}_{cyc}^s(s, s'') = \|s - s''\|_1, \tag{3}$$

where $s''$ is obtained from $s'$ via $F_{T \to S}$ and $A_T$, similarly to Equation 1.

This added loss makes our framework more robust in two ways: (1) it enforces the attended regions in the generated image to conserve content (e.g., pose), and (2) it encourages the attention maps to be sharp (converging towards a binary map), as the cycle-consistency loss of unattended areas will always be zero. Further, when computing $s''$, we use the attention map extracted from $A_T(s')$. This adds another consistency requirement, as the generated attention maps produced by $A_S$ and $A_T$ for $s$ and $s'$, respectively, should match to minimize Equation 3.

We obtain the final energy to optimize by combining the adversarial and cycle-consistency losses for both source and target domains:

$$\mathcal{L}(F_{S \to T}, F_{T \to S}, A_S, A_T, D_S, D_T) = \mathcal{L}_{adv}^s + \mathcal{L}_{adv}^t + \lambda_{cyc}\big(\mathcal{L}_{cyc}^s + \mathcal{L}_{cyc}^t\big), \tag{4}$$

where we use the loss hyper-parameter $\lambda_{cyc} = 10$ throughout our experiments. The optimal parameters of $\mathcal{L}$ are obtained by solving the minimax optimization problem:

$$F_{S \to T}^*, F_{T \to S}^*, A_S^*, A_T^*, D_S^*, D_T^* = \underset{F_{S \to T}, F_{T \to S}, A_S, A_T}{\operatorname{argmin}} \left( \underset{D_S, D_T}{\operatorname{argmax}} \ \mathcal{L}(F_{S \to T}, F_{T \to S}, A_S, A_T, D_S, D_T) \right). \tag{5}$$

## 3.2 Attention-guided discriminator

Equation 1 constrains the generators to act only on attended regions: as the attention networks train to become more accurate at finding the foreground, the generator improves in translating just the object of interest between domains, e.g., from horse to zebra. However, there is a tension: the whole-image discriminators look (implicitly) at the distribution of backgrounds with respect to the translated foregrounds. For instance, one observes that the translated horse now looks correctly like a zebra, but also that the overall scene is fake, because the background still shows where horses live—in meadows—and not where zebras live—in savannas. In this sense, we really are trying to make a 'fake' image which does not match either underlying probability distribution $P_S$ or $P_T$.

This tension manifests itself in two behaviors: (1) the generator $F_{S \rightarrow T}$ tries to 'paint' background directly into the attended regions, and (2) the attention map slowly includes more and more background, converging towards a fully attended map (all values in the map converge to 1). Our supplemental material provides example cases (last column in Figure 2; ablation studies Ours–D and Ours–D–A in Figure 5).

To overcome this, we train the discriminator such that it only considers attended regions. Simply using $s_a \odot s$ is problematic, as real samples fed to the discriminator now depend on the initially-untrained attention map $s_a$. This leads to mode collapse if all networks in the GAN are trained jointly. To overcome this issue, we first train the discriminators on full images for 30 epochs, and then switch to masked images once the attention networks $A_S$ and $A_T$ have developed.

Further, with a continuous attention map, the discriminator may receive 'fractional' pixel values, which may be close to zero early in training. While the generator benefits from being able to blend pixels at object boundaries, multiplying real images by these fractional values causes the discriminator to learn that mid gray is 'real' (i.e., we push the answer towards the midpoint 0 of the normalized $[-1,1]$ pixel space). Thus, we threshold the learned attention map for the discriminator:

$$t_{\text{new}} = \begin{cases} t & \text{if } A_T(t) > \tau \\ 0 & \text{otherwise} \end{cases} \quad \text{and} \quad s'_{\text{new}} = \begin{cases} F_{S \rightarrow T}(s) & \text{if } A_S(s) > \tau \\ 0 & \text{otherwise} \end{cases} \quad (6)$$

where $t_{\text{new}}$ and $s'_{\text{new}}$ are masked versions of target sample $t$ and translated source sample $s'$, which only contain pixels exceeding a user-defined attention threshold $\tau$, which we set to 0.1 (Figure 3 in the supplemental material justifies such choice). Moreover, we find that removing instance normalization from the discriminator at that stage is helpful as we do not want its final prediction to be influenced by zero values coming from the background.

Thus, we update the adversarial energy $\mathcal{L}_{adv}$ of Equation 2 to:

$$\mathcal{L}^s_{adv}(F_{S \rightarrow T}, A_S, D_T) = \mathbb{E}_{t \sim P_T(t)} \big[ \log(D_T(t_{\text{new}})) \big] + \mathbb{E}_{s \sim P_S(s)} \big[ \log(1 - D_T(s'_{\text{new}})) \big], \quad (7)$$

Algorithm 1 summarizes the training procedure for learning $F_{S \rightarrow T}$; training $F_{T \rightarrow S}$ is similar. Our supplemental material provides details of the individual network configurations.

When optimizing the objective in Equation 7 beyond 30 epochs, real image inputs to the discriminator are now also dependent on the learned attention maps. This can lead to mode collapse if the training is not performed carefully. For instance, if the mask returned by the attention network

---

**Algorithm 1** Training procedure for the source-to-target map $F_{S \rightarrow T}$.

**Input:** $\mathcal{X}_S$, $\mathcal{X}_T$, $K$ (number of epochs), $\lambda_{cyc}$ (cycle-consistency weight), $\alpha$ (ADAM learning rate).
1: **for** $c = 0$ to $K-1$ **do**
2:     **for** $i = 0$ to $|\mathcal{X}_S| - 1$ **do**
3:         Sample a data point $s$ from $\mathcal{X}_S$ and a data point $t$ from $\mathcal{X}_T$.
4:         **if** $c < 30$ **then**
5:             Compute $s'$ using Equation 1.
6:             Update parameters of $F_{S \rightarrow T}$, $D_T$, and $A_S$ using Equation 4 with learning rate $\alpha$.
7:         **else**
8:             Compute $s'_{new}$ and $t_{new}$ using Equation 6.
9:             Update parameters of $F_{S \rightarrow T}$ and $D_T$ using Equations 4 and 7 with learning rate $\alpha$.
10:         **end if**
11:     **end for**
12: **end for**
**Output:** Trained networks $F^*_{S \rightarrow T}$, $A^*_S$ and $D^*_T$.

---

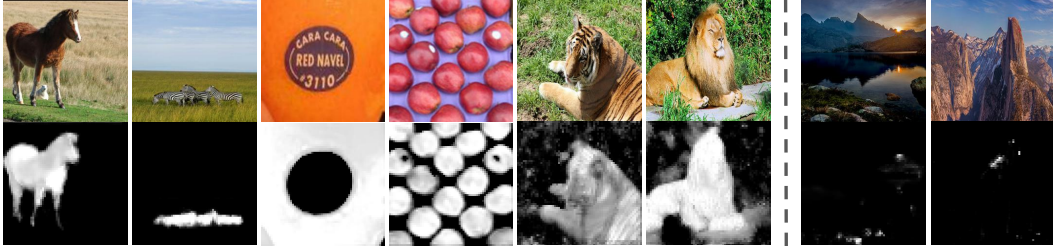

Figure 3: Input source images (top row) and their corresponding estimated attention maps (below). These reflect the discriminative areas between the source and target domains. The right side of the figure shows source and target attention maps, trained on horses and zebras, respectively, when applied to images without horse or zebra. The lack of attention suggests appropriate attention network behavior.

| Input | Our Attention | Ours | CycleGAN [1] | RA [2] | DiscoGAN [3] | UNIT [4] | DualGAN [5] |
|---|---|---|---|---|---|---|---|

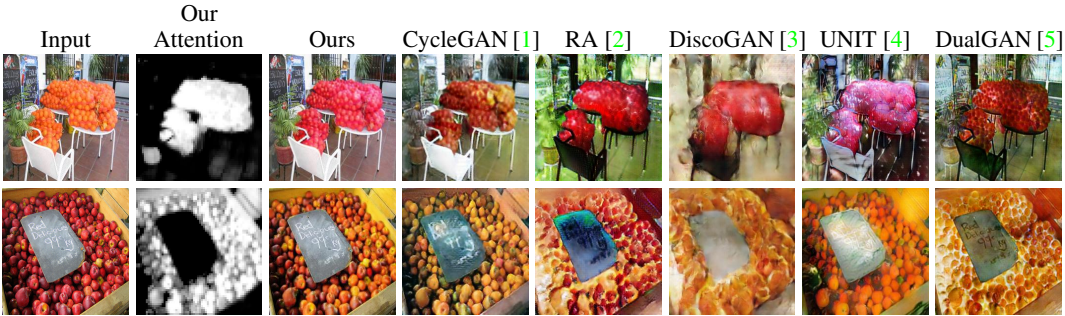

Figure 4: Image translation results for mapping apples to oranges and our learned attention.

is always zero, then the generator will always create 'real' images from the point of view of the discriminator, as the masked sample $t_{\text{new}}$ in Equation 7 would be all black. We avoid this situation by stopping the training of both $A_S$ and $A_T$ after 30 epochs (Figure 2 in the supplementary material justifies such hyper-parameter choice).

## 4   Experiments

**Baselines.**   We compare to DiscoGAN [3] and CycleGAN [1], which are similar, but which use different losses: DiscoGAN uses a standard GAN loss [12], and CycleGAN uses a least-squared GAN loss [26]. We also compare with DualGAN [5], which is similar to CycleGAN but uses a Wasserstein GAN loss [27]. Aditionally, we compare with Liu et al.'s UNIT algorithm [4], which leverages the latent space assumption between each pair of source/target images. Finally, we compare with Wang et al.'s attention module [2] by incorporating it after the first layer of our generators; we refer to this implementation as "RA".

**Datasets.**   We use the 'Apple to Orange' ($A \leftrightarrow O$) and 'Horse to Zebra' ($H \leftrightarrow Z$) datasets provided by Zhu et al. [1], and the 'Lion to Tiger' ($L \leftrightarrow T$) dataset obtained from the corresponding classes in the Animals With Attributes (AWA) dataset [28]. These datasets contain objects at different scales across different backgrounds, which make the image-to-image translation setting more challenging. Note that for the mapping Lion to Tiger we do not find it necessary to apply the attention-guided discriminator part.

**Qualitative results.**   Observing our learned attention maps, we can see that our approach is able to learn relevant image regions and ignore the background (Figure 3). When an input image does not contain any elements of the source domain, our approach does not attend to it, and so successfully leaves the image unedited. Holistic image translation approaches, on the other hand, are mislead by irrelevant background content and so incorrectly hallucinate texture patterns of the target objects (last two rows of Figure 5).

Among competing approaches, DiscoGAN struggles to separate the background and foreground content (see Figures 1, 4 and 5). We believe this is partly because their cycle-consistency energy is given the same weight as the GAN's adversarial energy. DualGAN produces slightly better results,

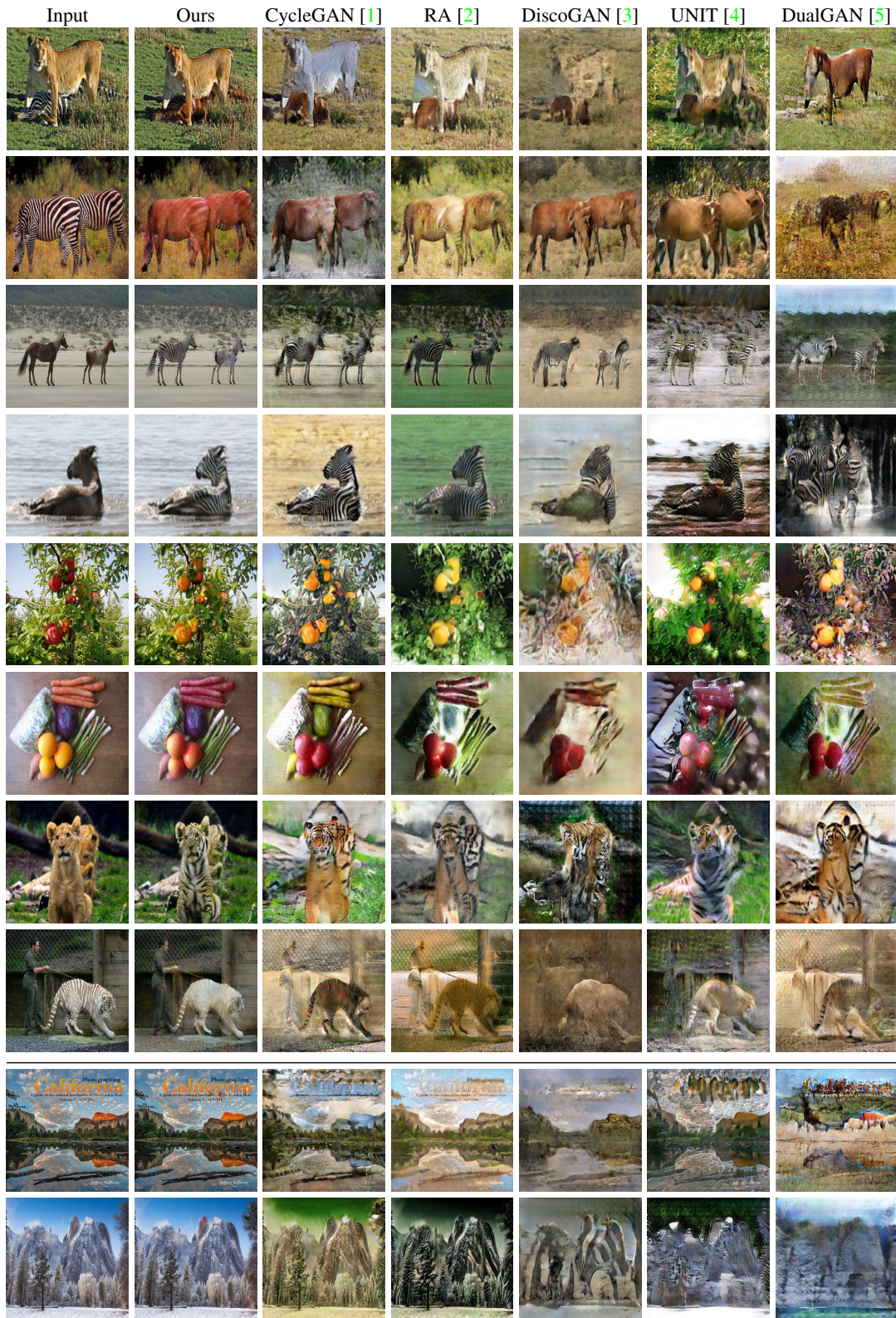

Figure 5: Translation results. From top to bottom: $Z \to H$, $Z \to H$, $H \to Z$, $H \to Z$, $A \to O$, $O \to A$, $L \to T$, and $T \to L$. Below line: image translation in the absence of the source domain class ($Z \to H$).

Table 1: Kernel Inception Distance$\times 100 \pm$ std.$\times 100$ for different image translation algorithms. Lower is better. Abbreviations: ($A$)pple, ($O$)range, ($H$)orse, ($Z$)ebra, ($T$)iger, ($L$)ion.

| Algorithm | $A \to O$ | $O \to A$ | $Z \to H$ | $H \to Z$ | $L \to T$ | $T \to L$ |
|---|---|---|---|---|---|---|
| DiscoGAN [3] | $18.34 \pm 0.75$ | $21.56 \pm 0.80$ | $16.60 \pm 0.50$ | $13.68 \pm 0.28$ | $16.10 \pm 0.55$ | $19.97 \pm 0.09$ |
| RA [2] | $12.75 \pm 0.49$ | $13.84 \pm 0.78$ | $10.97 \pm 0.26$ | $10.16 \pm 0.12$ | $9.98 \pm 0.13$ | $12.68 \pm 0.07$ |
| DualGAN [5] | $13.04 \pm 0.72$ | $12.42 \pm 0.88$ | $12.86 \pm 0.50$ | $10.38 \pm 0.31$ | $10.18 \pm 0.15$ | $10.44 \pm 0.04$ |
| UNIT [4] | $11.68 \pm 0.43$ | $11.76 \pm 0.51$ | $13.63 \pm 0.34$ | $11.22 \pm 0.24$ | $11.00 \pm 0.09$ | $10.23 \pm 0.03$ |
| CycleGAN [1] | $8.48 \pm 0.53$ | $9.82 \pm 0.51$ | $11.44 \pm 0.38$ | $10.25 \pm 0.25$ | $10.15 \pm 0.08$ | $10.97 \pm 0.04$ |
| Ours | $\mathbf{6.44 \pm 0.69}$ | $\mathbf{5.32 \pm 0.48}$ | $\mathbf{8.87 \pm 0.26}$ | $\mathbf{6.93 \pm 0.27}$ | $\mathbf{8.56 \pm 0.16}$ | $\mathbf{9.17 \pm 0.07}$ |

although the background is still heavily altered. For example, the first row of Figure 1 contains undesirable zebra patterns in the background. CycleGAN produces more visually appealing results with its least-squares GAN and appropriate weighting between the adversarial and cycle-consistency losses, even though some elements of the background are still altered. For instance, CycleGAN alters the writing on the chalkboard in the last row of Figure 4, and generates a blue-grey lion in the first row of Figure 5 when asked to translate the zebra pinned down by the lion. The UNIT algorithm uses the shared latent space assumption between source and target domains to be robust to changes in geometric shape. For example, in the 7th row of Figure 5, we can see that the face of the lion cub is mapped to a tiger; however, the overall image is not realistic. Finally, incorporating residual attention (RA) modules into the image translation framework does not improve the generated image quality, which validates our choice of incorporating attention into images instead of on activation functions. This is particularly noticeable when the input source image does not contain any relevant object, as in Figure 5 (bottom). In this case, existing algorithms are mislead by irrelevant background content and incorrectly hallucinate texture patterns of the target objects. By learning attention maps, our algorithm successfully ignores background contents and reproduces the input images.

One limitation of our approach is visible in the last third row of Figure 5, which contains an albino tiger. In this challenging case of an object with outlier appearance within its domain, our attention network fails to identify the tiger as foreground, and so our network changes the background image content, too. However, overall, our approach of learning attention maps within unsupervised image-to-image translation obtains more realistic results, particularly for datasets containing objects at multiple scales and with different backgrounds.

**Quantitative results.** We use the recently proposed Kernel Inception Distance (KID) [29] to quantitatively evaluate our image translation framework. KID computes the squared maximum mean discrepancy (MMD) between feature representations of real and generated images. Such feature representations are extracted from the Inception network architecture [30]. In contrast to the Fréchet Inception Distance [31], KID has an unbiased estimator, which makes it more reliable, especially when there are fewer test images than the dimensionality of the inception features. While KID is not bounded, the lower its value, the more shared visual similarities there are between real and generated images. As we wish the foreground of mapped images to be in the target domain $T$ and the background to remain in the source domain $S$, a good mapping should have a low KID value when computed using both the target and the source domains. Therefore, we report the mean KID value computed between generated samples using both source and target domains in Table 1. Further, to ensure consistency, the mean KID values reported are averaged over 10 different splits of size 50, randomly sampled from each domain.

Our approach achieves the lowest KID score in all the mappings, with CycleGAN as the next best performing approach. UNIT achieves the second-lowest KID score, which suggests that the latent space assumption is useful in our setting. Using Wasserstein GAN allows DualGAN to follow closely behind. The CycleGAN variant using residual attention modules (RA) produces worse results than regular CycleGAN but comparable to UNIT, which suggests that applying attention on the feature space does not considerably improve performance. Finally, by giving the same weight to the adversarial and cyclic energy, DiscoGAN achieves the worst performance in terms of mean KID values, which is consistent with our qualitative results.

**Ablation Study.** First, we evaluate the cycle-consistency loss governed by Equation 3. This is motivated by using attention to constrain the mapping between only relevant instances, which can be considered as a weak form of cycle consistency. The cycle-consistency loss plays an important role in making attention maps sharp; without them, we notice an onset of mode collapse in GAN training. As a result, we obtain a model ('Ours–cycle') with very high KID (Table 2).

Table 2: Kernel Inception Distance$\times 100 \pm$ std.$\times 100$ for ablations of our algorithm. Lower is better. Abbreviations: $(H)$orse, $(Z)$ebra.

| Algorithm | $Z \rightarrow H$ | $H \rightarrow Z$ |
|---|---|---|
| Ours–cycle | $64.55 \pm 0.34$ | $41.48 \pm 0.34$ |
| Ours–cycleAtt | $9.46 \pm 0.38$ | $7.79 \pm 0.23$ |
| Ours–As | $10.90 \pm 0.25$ | $7.62 \pm 0.25$ |
| Ours–At | $9.30 \pm 0.45$ | $7.80 \pm 0.21$ |
| Ours–D | $9.26 \pm 0.22$ | $7.77 \pm 0.35$ |
| Ours–D–A | $9.86 \pm 0.32$ | $8.28 \pm 0.34$ |
| Ours | $\mathbf{8.87 \pm 0.26}$ | $\mathbf{6.93 \pm 0.27}$ |

Next, we test the effect of computing attention on the inverse mapping. Instead of computing a new attention map $A_T(s')$, we use the formerly computed $A_S(s)$. This model ('Ours–cycleAtt') performs worse, because computing attention on both the mapping and its inverse indirectly enforces similarity between both attention maps $A_T(s')$ and $A_S(s)$.

Further, we evaluate behavior with only a single attention network: 'Ours–As' and 'Ours–At' corresponding to $A_S$ and $A_T$, respectively. These approaches are the best performing after our final implementation: $A_S$ acts on $s$, but also on $t'$ via the inverse mapping, which influences the generators to still only translate relevant regions. Moreover, we measure the importance of our attention-guided discriminator by replacing it with a whole-image discriminator while stopping the training of the attention networks ('Ours–D'). For this model, mean KID values are higher than our final formulation because the generator tries to paint elements of the background onto the foreground to compensate for the variance between foreground and background in the source and target domains.

Finally, we consider the contemporaneous Attention GAN of Chen et al. [25], which also learns an attention map for image translation through a cyclic loss. We compare their approach using an ablated version of our software implementation, as we await a code release from the authors for a direct results comparison. Our approach differs in two ways: first, we feed the holistic image to the discriminator for the first 30 epochs, and afterwards show it only the masked image; second, we stop the training of the attention networks after 30 epochs to prevent it from focusing on the background as well. These two differences reduce errors caused by spurious image additions from $F$, and remove the need for the optional supervision introduced by Chen et al. to help remove background artifacts and better 'focus' the attention map on the foreground. Table 2 demonstrates this quantitatively ('Ours–D–A'), with higher KID scores compared to our final implementation. Please see the supplemental document for visual examples.

## 5  Conclusion

While recent unsupervised image-to-image translation techniques are able to map relevant image regions, they also inadvertently map irrelevant regions, too. By doing so, the generated images fail to look realistic, as the background and foreground are generally not blended properly. By incorporating an attention mechanism into unsupervised image-to-image translation, we demonstrate significant improvements in the quality of generated images. Our simple algorithm leverages the discriminator to learn accurate attention maps with no additional supervision. This suggests that our learned attention maps reflect where the discriminator looks before deciding whether an image is real or fake, making it an appropriate tool for investigating the behavior of adversarial networks.

**Future work.**  Although our approach can produce appealing translation results in the presence of multi-scale objects and varying backgrounds, the overall approach is still not robust to shape changes between domains, e.g., making Pegasus by translating a horse into a bird. Our transfer must happen within attended regions in the image, but shape change typically requires altering parts outside these regions. In the supplementary material, we provide an example of such limitation via the mapping zebra to lion. Our code is released in the following Github repository: https://github.com/AlamiMejjati/Unsupervised-Attention-guided-Image-to-Image-Translation.

**Acknowledgements:**  Youssef A. Mejjati thanks the Marie Sklodowska-Curie grant agreement No 665992, and the UK's EPSRC Centre for Doctoral Training in Digital Entertainment (CDE), EP/L016540/1. Kwang In Kim, Christian Richardt, and Darren Cosker thank RCUK EP/M023281/1.

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
