[Supplementary Material · 1862-supplemental.pdf]

# Supplemental Document for Unsupervised Attention-guided Image-to-Image Translation

**Youssef A. Mejjati**
University of Bath

**Christian Richardt**
University of Bath

**James Tompkin**
Brown University

**Darren Cosker**
University of Bath

**Kwang In Kim**
University of Bath

We provide details of the network architectures we used for each of the generator, discriminator, and attention networks in Section 1, then we discuss limitations of our approach and show results of our algorithm when translation is needed across the whole image (e.g., summer to winter). In addition, we discuss hyper-parameter selection in Section 3. Finally, we provide additional example results for qualitative comparisons in Section 4.

## 1 Network architecture

**Generators** $F_{S \to T}$ **and** $F_{T \to S}$**.** Our generator architecture is similar to the CycleGAN generator [1]. Adopting CycleGAN's notation, "`c7s1-`$k$`-R`" denotes a 7×7 convolution with stride 1 and $k$ filters, followed by a ReLU activation ('`R`'). "`tc`$k$`s2`" denotes a 3×3 transpose convolution operation (sometimes called 'deconvolution') with $k$ filters and stride 2, followed by a ReLU activation. "`r`$k$" denotes a residual block formed by two 3×3 convolutions with $k$ filters, stride 1 and a ReLU activation. Sigmoid activation is indicated by '`S`' and 'tanh' by '`T`'. We apply Instance normalization after all layers apart from the last layer.

Our generator architecture is: `c7s1-32-R, c3s2-64-R, c3s2-128-R, r128, r128, r128, r128, r128, r128, r128, r128, r128, tc64s2, tc32s2, c3s1-3-T`.

**Attention networks** $A_S$ **and** $A_T$**.** In our attention networks, we use instance normalization in all layers apart from the last layer. Further, instead of using transpose convolutions, we use nearest-neighbor upsampling layers "`up2`" that doubles the height and width of its input. We follow the upsampling layers with 3×3 convolutions of stride 1 with ReLU activations, apart from the last layer, which uses a sigmoid.

Our attention network architecture is:
`c7s1-32-R, c3s2-64-R, r64, up2, c3s1-64-R, up2, c3s1-32-R, c7s1-1-S`.

**Discriminators** $D_S$ **and** $D_T$**.** We adopt the CycleGAN discriminator architecture: We use instance normalization everywhere apart from the last layer. However, when we start feeding only the foreground to the discriminator (after 30 epochs), we remove instance normalization as the input is at this stage a masked image and we do not want zero values to influence the generation process. In addition, instead of ReLUs, we use Leaky-ReLUs (LR) with slope 0.2.

Our discriminator architecture is:
`c4s2-64-LR, c4s2-128-LR, c4s2-256-LR, c4s1-512-LR, c4s1-1`.

Finally, similar to CycleGAN we adopt Least Square GAN (LSGAN), as we find that it helps producing sharper images.

## 2    Limitation of our approach

Although our approach can produce appealing translation results in the presence of multi-scale objects and varying backgrounds, the overall approach is still not robust to shape changes between domains, e.g., mapping zebras to lions depicted in Figure 1. Our transfer must happen within attended regions in the image, but shape change typically requires altering parts outside these regions. Consequently, the attention maps end up covering areas in the background in order to allow for this geometric change, however similar to CycleGAN such changes are limited due to the cyclic consistency constraint.

Figure 1: A limitation of our algorithm is its lack of robustness to significant geometric changes as illustrated by the Lion $\rightarrow$ Zebra Mapping (left), and Zebra $\rightarrow$ Lion Mapping (right).

## 3    Hyper-parameter tuning

Our algorithm is characterized by two training stages. In the first stage we train $F_S$, $F_T$, $A_S$, $A_T$ and both discriminators $D_S$ and $D_T$. In addition, the discriminators are trained with the holistic images as input. In the second stage we interrupt the training of attention networks $A_S$ and $A_T$ and train the discriminators using the foregrounds only. We apply this strategy as we noticed that when carrying the training using only the first training stage, the attention maps are also focused on the background. Such behavior is explained by different background scenes covering horse and zebra images (the first lives in green meadows, while the former lives in dry Savannah landscapes). Figure 2 depicts such behavior: as the switching epoch between the first and second stage increases, more and more of the background is included in the attention maps (last columns in Figure 2); on the other hand, if the switching point is done too early then the attention map fail to cover the entire foreground (first column in Figure 2).

Before feeding the foreground to the discriminator in the second stage, we threshold the attention masks to make them binary. This avoids feeding fractional values to the discriminators which stops it from learning that mid gray values in the foreground are real, making the generation process more realistic. Figure 3 shows the effect of varying the threshold $\tau$ on the generated images: Low values of $\tau$ give equivalent results as the background in the learned attention images tend to be close to zero; however the higher $\tau$ gets, the less realistic are the generated images as foreground areas with lower uncertainty (e.g. due to unusual illumination or pose) are not taken into account in such scenario. This is specially the case for the mapping horse $\rightarrow$ zebra.

## 4    Additional results

Figure 4 shows mapping results when the image requires holistic changes, here summer to winter and winter to summer. Even though our algorithm is not initially designed for such use case, we found that it is able to create attention maps focusing on the entire image. Note that since there is no clear distinction between foreground and background in this scenario, we do not apply Eq. 7 in this particular mapping. Further, this scenario required a longer training time (200 vs. 100 epochs).

Figure 5 shows qualitative transfer results for our ablation experiments.

Figures 6 to 12 show example translation results for qualitative evaluation across six datasets, plus an example on domains which do not contain the object of interest (Figure 12).

|        | Input | 10 epochs | 30 epochs | 50 epochs | 70 epochs |
|--------|-------|-----------|-----------|-----------|-----------|

Figure 2: Effect of varying the number of epochs before stopping the training of the attention networks and replacing the input to the discriminator with the foreground only.

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

| Input | 0.1 | 0.3 | 0.5 | 0.7 |
|-------|-----|-----|-----|-----|

Figure 3: Effect of varying the threshold parameter $\tau$. Low threshold values give similar results while higher values results on less realistic mappings.

| Input | Our attention map | Generated | Input | Our attention map | Generated |
|-------|-------------------|-----------|-------|-------------------|-----------|

Figure 4: Even in presence of images requiring holistic changes, our algorithm is able to producing attention maps focusing on the entire image, which results on good quality generated images. *Left:* Summer to Winter mapping, *Right:* Winter to summer mapping.

| Input | Ours | Ours–As | Ours–At | Ours–cycleAtt | Ours–D | Ours–D–A | Ours–cycle |
|-------|------|---------|---------|---------------|--------|----------|------------|

Figure 5: Qualitative results for our ablation experiments. The images produced by our final formulation are sharper and more realistic compared to other approaches. Specifically, removing the cycle-consistency loss ('Ours–cycle') leads to the collapse of the GAN training, confirming its essential role in the image translation framework. By only adopting the holistic image discriminator ('Ours–D') and not stopping the training of the attention networks (Ours–D–A), we notice artifacts on the foreground and background as shown in the second and bottom row respectively. Furthermore, removing one attention network from our algorithm ('Ours–As', 'Ours–At') degrades the visibly quality of the generated images. Even though the quality decreases in this case, only the foreground in the generated images gets altered, which is an interesting observation even though one of the cycles ($S \rightarrow T$ or $T \rightarrow S$) lack an attention estimator. Finally, reusing attention on the cycle pass of our algorithm ('Ours–cycleAtt') results on less sharp images compared to our final implementation.

Input     Ours     CycleGAN [1]     RA [2]     DiscoGAN [3]     UNIT [4]     DualGAN [5]

Figure 6: Zebra → Horse translation results.

| Input | Ours | CycleGAN [1] | RA [2] | DiscoGAN [3] | UNIT [4] | DualGAN [5] |
|-------|------|--------------|--------|--------------|----------|-------------|

Figure 7: Horse → Zebra translation results.

| Input | Ours | CycleGAN [1] | RA [2] | DiscoGAN [3] | UNIT [4] | DualGAN [5] |
|-------|------|--------------|--------|--------------|----------|-------------|

Figure 8: Apple → Orange translation results.

Figure 9: Orange → Apple translation results.

| Input | Ours | CycleGAN [1] | RA [2] | DiscoGAN [3] | UNIT [4] | DualGAN [5] |
|-------|------|--------------|--------|--------------|----------|-------------|

Figure 10: Lion → Tiger translation results.

| Input | Ours | CycleGAN [1] | RA [2] | DiscoGAN [3] | UNIT [4] | DualGAN [5] |

Figure 11: Tiger → Lion translation results.

| Input | Ours | CycleGAN [1] | RA [2] | DiscoGAN [3] | UNIT [4] | DualGAN [5] |
|-------|------|--------------|--------|--------------|----------|-------------|

Figure 12: Image translation of Horse → Zebra on images without horses or zebras. By explicitly estimating attention maps, our algorithm successfully *ignores* irrelevant backgrounds and correctly reproduces the input images. Existing algorithms are mislead by these backgrounds and incorrectly hallucinate zebra stripe patterns.