[Reviews · NeurIPS 2018]

Reviewer 1



To conduct image-to-image translation, this paper introduces attention mechanisms to CycleGAN [1]. The divergence between only the relevant parts of both source and target domains is minimized. Compared to recent approaches, the proposed method performs better in conserving the background and gets the lowest FID in quantitative analysis. Related works Related works follows a clear structure. For image-to-image translation, it introduces supervised and unsupervised methods, followed by mentioning the problem which is the background is affected, and then it refers to some related research on attention learning. Approach The idea of having continuous attention map for generator is good, and it is also good to read the authors’ explanations of why and how it is valuable. The reason why they use segmented attention map for discriminator is also well presented. Experiemnts It is interesting to see Figure 3 and Figure 5 have translation results when there is no interested class in source domain, showing this approach applies appropriate attention mechanisms. Some experiment details like the choices of hyperparameters are absent, not being presented in the submission paper or supplementary material. But in the end of the paper, the authors say the code will be released, hopefully this will be the case. Lack of qualitative analysis about UNIT approach[4]. Other comments The readability of this paper is good. The structure and diagram of the proposed attention method are clearly presented. In conclusion, the authors mention one constraint of the proposed method, which is not being robust to shape changes, but this paper doesn’t involve related examples. It would be better to have the mentioned horse-to-bird results in supplementary material. L210: the last row of Figure 5 doesn’t seem to be an albino tiger, but the last third row is, which the last row of the top part.

Reviewer 2



The authors of this manuscript propose an attention strategy in GAN for image-to-image translation. Their experiments show that the proposed method can well generate better objects without changing the background. Compared with existing GAN method, it is clearly a significant improvement.

Reviewer 3



This paper proposed an attention based model for unsupervised image-to-image translation. The attention mechanism is employed to untangle the image's foreground objects from the background, which is novel the interesting. The quality of generated images is very good and better than previous work, demonstrating the effectiveness of this model on several datasets. One limitation of the proposed model, besides the discussion in future work part, is that such attention model can only be applied to the images includes clear and separable foreground and background, e,g., zebra<->horse. It requires the images from source domain and images from target domain have some overlapping region (i.e. background) so that it cannot be applied to many problems such as style translation and colorization or many other datasets such as day <-> night, Google maps <-> aerial photos, labels <-> facades and labels <-> cityscapes. The experiments do not provide the influence of many hyperparameters, such as the number of epochs trained before applying (7) and the threshold in (6). Such discussion can make the experiments more convincing. This paper missed the discussion of an important reference "Generating Videos with Scene Dynamics". They also learned a binary mask to learn the static background and dynamic foreground in video. Although it is not called "attention", I think this paper share the same spirit.